# Multimodal Tracking of Hematopoietic Stem Cells from Young and Old Mice Labeled with Magnetic–Fluorescent Nanoparticles and Their Grafting by Bioluminescence in a Bone Marrow Transplant Model

**DOI:** 10.3390/biomedicines9070752

**Published:** 2021-06-29

**Authors:** Fernando A. Oliveira, Mariana P. Nucci, Javier B. Mamani, Arielly H. Alves, Gabriel N. A. Rego, Andrea T. Kondo, Nelson Hamerschlak, Mara S. Junqueira, Lucas E. B. de Souza, Lionel F. Gamarra

**Affiliations:** 1Hospital Israelita Albert Einstein, São Paulo 05652-000, SP, Brazil; fernando.anselmo@einstein.br (F.A.O.); mariana.nucci@hc.fm.usp.br (M.P.N.); javierbm@einstein.br (J.B.M.); arielly.alves@einstein.br (A.H.A.); gabriel.nery@einstein.br (G.N.A.R.); andrea.kondo@einstein.br (A.T.K.); hamer@einstein.br (N.H.); 2LIM44—Hospital das Clínicas da Faculdade Medicina da Universidade de São Paulo, São Paulo 05403-000, SP, Brazil; 3Center for Translational Research in Oncology, Cancer Institute of the State of Sao Paulo—ICESP, São Paulo 01246-000, SP, Brazil; mara.junqueira@hc.fm.usp.br; 4Hemocentro de Ribeirão Preto, Faculdade de Medicina de Ribeirão Preto, Universidade de São Paulo, Ribeirão Preto 14051-060, SP, Brazil; lucasebsouza@usp.br

**Keywords:** hematopoietic stem cell, nanoparticle, homing, tracking, near-infrared fluorescence imaging, magnetic resonance imaging, bioluminescence, molecular imaging, noninvasive imaging

## Abstract

This study proposes an innovative way to evaluate the homing and tracking of hematopoietic stem cells from young and old mice labeled with SPION_NIRF-Rh_ conjugated with two types of fluorophores (NIRF and Rhodamine), and their grafting by bioluminescence (BLI) in a bone marrow transplant (BMT) model. In an in vitro study, we isolated bone marrow mononuclear cells (BM-MNC) from young and old mice, and analyzed the physical–chemical characteristics of SPION_NIRF-Rh_, their internalization, cell viability, and the iron quantification by NIRF, ICP-MS, and MRI. The in vivo study was performed in a BMT model to evaluate the homing, tracking, and grafting of young and old BM-MNC labeled with SPION_NIRF-Rh_ by NIRF and BLI, as well as the hematological reconstitution for 120 days. 5FU influenced the number of cells isolated mainly in young cells. SPION_NIRF-Rh_ had adequate characteristics for efficient internalization into BM-MNC. The iron load quantification by NIRF, ICP-MS, and MRI was in the order of 10^4^ SPION_NIRF-Rh_/BM-MNC. In the in vivo study, the acute NIRF evaluation showed higher signal intensity in the spinal cord and abdominal region, and the BLI evaluation allowed follow-up (11–120 days), achieving a peak of intensity at 30 days, which remained stable around 10^8^ photons/s until the end. The hematologic evaluation showed similar behavior until 30 days and the histological results confirm that iron is present in almost all tissue evaluated. Our results on BM-MNC homing and tracking in the BMT model did not show a difference in migration or grafting of cells from young or old mice, with the hemogram analysis trending to differentiation towards the myeloid lineage in mice that received cells from old animals. The cell homing by NIRF and long term cell follow-up by BLI highlighted the relevance of the multimodal nanoparticles and combined techniques for evaluation.

## 1. Introduction

Bone marrow transplantation (BMT) is a therapeutic procedure used as a treatment for a number of hematological diseases, and also for some non-hematological diseases [1,2]. It is traditionally performed by the intravenous infusion of hematopoietic stem cell (HSC) into a recipient after performing a conditioning regimen to destroy the defective spinal cord. Transplanted HSC can be isolated from some sources such as bone marrow (BM), umbilical cord blood or peripheral blood, and is intended to restore the proliferative function of the recipient marrow [3,4,5]. Once BMT is performed, the hope is that the cells administered in the circulation will find their way home and repopulate the recipient’s marrow. This cell migration process to hematopoietic niches localized into bone marrow is known as homing [6]; this active process begins after the HSC administration and does not last more than 48 h [7,8]. However, not all transplanted HSC end up reaching the hematopoietic niches; only some of the cells are able to reach the site of interest and reestablish the function of BM. The remainder of the transplanted cells ends up being retained when passing through non-hematopoietic organs, failing to reach the place of interest in BM [9]. Aging is another prejudicial factor for homing; transplanted HSC from old donors has a lower capacity to migrate and graft than cells from young donors [10,11]. In addition, the graft does not have the same quality, resulting in increased cell proliferation, reduced expansion, and accumulation of DNA damage, among others [12,13].

Generally, studies about cells homing/tracking after BTM have used ex vivo techniques to quantify HSC in recipient animal; this type of analysis cannot generate data on the migration or grafting dynamic in a single animal longitudinally, since it requires euthanasia at the point of analysis [14,15,16]. However, the cell homing and grafting data can be obtained in a non-invasive way with the use of the molecular image technique [17]. The molecular image comprise a group of image techniques that use the target molecules to track biological processes via in vivo analysis, in a non-invasive way, besides allowing the quantitative analysis [18]. Bioluminescence Imaging (BLI) is the most commonly used technique for non-invasive HSC tracking in experimental models [17]. We can also highlight techniques such as Magnetic Resonance Imaging (MRI), Positron Emission Tomography (PET), Single Photon Emission Computed Tomography (SPECT) and Fluorescence Imaging (FLI), and especially Near-infrared fluorescence (NIRF) that enables the capture of images with greater depth of penetration than fluorescent agents in the visible spectrum, reducing the autofluorescence of tissues [19]. Each of these image modalities has advantages and limitations when compared to any other, which is why the development of probes which make it possible to perform a set of images with a single tracking agent has gained a lot of attention due to the multimodal images generated [17,20]. The last systematic review on this topic, Noninvasive Tracking of Hematopoietic Stem Cells in a Bone Marrow Transplant Model, did not show the combined use of the molecular image technique in any of the studies analyzed [17].

This study proposes an innovative way to evaluate the homing and tracking of hematopoietic stem cells from young and old mice labeled with Superparamagnetic Iron Oxide Nanoparticles conjugated with two types of fluorophores (NIRF and Rhodamine) (SPION_NIRF-Rh_), providing magnetic and fluorescence properties, and their grafting by BLI in a BMT model. This multimodal approach aims to minimize the limitations of each technique and seek more robust data on the migration and grafting of young and old cells in a BMT model.

## 2. Materials and Methods

### 2.1. Ethical Aspects in the Use of Animals

The study was approved by the Ethics in Animal Research Committee of the *Hospital Israelita Albert Einstein* (Sao Paulo, Brazil), number 3133/17. The animals were maintained at the vivarium of the *Centro de Experimentação e Treinamento em Cirurgia* (CETEC) of the *Instituto Israelita de Ensino e Pesquisa Albert Einstein* (IIEP) in the following conditions: at 21 ± 2 °C and 60% ± 5% relative humidity with full ventilation, under a 12 h light/dark cycle (7 a.m.–7 p.m.), and they had access to food and water ad libitum. This vivarium is accredited by the Association for the Assessment and Accreditation of Laboratory Animal Care International (AAALAC International), and the general conditions were monitored daily.

Male and female C57BL/6 mice, ages ranging 6 to 8 weeks (young mice) and more than 18 months (old mice) were used in this study.

### 2.2. In Vitro Study

#### 2.2.1. Extraction and Isolation of Bone Marrow Mononuclear Cells (BM-MNC)

BM-MNC were isolated by density gradient on Ficoll-Paque™ Premium (1.084 g/mL) (GE Healthcare, Uppsala, Sweden). Four days before cell extraction we administered 150 mg/kg of 5-fluorouracil (5-FU) (EMD Millipore Corp., St. Louis, MO, USA) i.p. in young and old C57BL/6 mice, aiming at enriching the hematopoietic stem cell (HSC) compartment of the BM-MNC pool. Mice were euthanized by an overdose of anesthesia, their femurs and tibias were extracted and cleaned with 70% of alcohol and phosphate-buffered saline (PBS), and centrifuged inside of a tip allocated in a conical tube at 1000× *g* for 5 min at 21 °C. After centrifugation, the cell pellet was collected and resuspended in 4 mL of StemSpan Serum-Free Expansion Medium (SFEM; Stem Cell Technologies, Vancouver, Canada), supplemented with 100 U/mL streptomycins and 100 U/mL penicillin (GIBCO—Invitrogen Technologies, Grand Island, New York, NY, USA), interleukin 3 (IL-3) (10 ng/mL), interleukin 6 (IL-6) (10 ng/mL), and stem cell factor (SCF) (100 ng/mL) (PeproTech, Rocky Hill, NJ, USA) (referred to as complete SFEM), centrifuged at 500× *g* for 5 min at 21 °C. The pellet was resuspended again in 4 mL of SFEM, and it was added carefully above 4mL of Ficoll, without mixing, in another conical tube that was centrifuged at 400× *g* for 30 min at 21 °C, without a break. After centrifugation, the halo of BM-MNC formed between medium and the Ficoll fraction was aspirated, and the cells were washed with SFEM, counted with the Neubauer chamber with the help of the Trypan blue stain, and after counting the cells were seeded at 1 × 10^6^ cells/mL in 24-well plates containing SFEM supplemented with hematopoietic cytokines and incubated at 37 °C with 5% de CO_2_.

#### 2.2.2. Immunophenotypic Characteristics of BM-MNC

The cell characterization of lineage-Sca-1+ c-Kit+ (LKS) present in BM-MNC isolated from young and old C57BL/6 mice was performed by flow cytometry (FACSAria™ III, BD Biosciences, Franklin Lakes, NJ, USA), using the BD Mouse Hematopoietic Stem and Progenitor Cell Isolation Kit (BD Biosciences, NJ, USA) that contains specific immunophenotypic markers for isolation of HSC and HPC as Sca-1 and c-Kit (CD177), and also markers of hematopoietic lineages such as CD3 (T cell marker), CD45R (B cell marker), Ly6C and Ly6G (granulocyte markers), CD11b (macrophage marker), and TER-119 (erythrocyte marker).

For the LKS cells’ separation, BM-MNC isolation of young and old mice with and without the administration of 5-FU (4 days before cell isolation) was performed previously; these cells were incubated with the following antibodies to HSC and to the hematopoietic lineage: PE c-Kit, PE-Cy7 Sca-1, the cocktail of antigen-presenting cell markers (CAA) and 7-Amino-Actinomycin D (7-AAD) for cell viability. The cells were selected first for viability, then for size and granularity, lineage markers, and finally for HSC markers. The data obtained were analyzed using the FACSDIVA software (BD Biosciences, San Jose, CA, USA) and FlowJo.

#### 2.2.3. BM-MNC Lentiviral Transduction for Luciferase Expressing

BM-MNC were transduced with viruses carrying the lentiviral vector pMSCV-Luc2-T2A-Puro which codifies a codon-optimized version of luciferase. For transduction, BM-MNC previously cultured for 48 h in complete SFEM were mixed with lentivirus particles (5 virus/cell) in the presence of 8 µg/mL polybrene (Sigma-Aldrich, St.Louis, MO, USA) and centrifuged at 1000× *g* for 90 min at 33 °C in a 50 mL conical tube. After centrifugation, cells were transferred to a 24-well plate and incubated overnight. The next day, a new transduction cycle was performed, as described above.

#### 2.2.4. Efficacy Evaluation of Luciferase Transduction in BM-MNC by Bioluminescent Imaging (BLI)

For quantification of bioluminescence, 2 × 10^5^, 1 × 10^5^, 5 × 10^4^, and 2.5 × 10^4^ transduced BM-MNC were seeded in quadruplicates in a 96-well black culture plate, Next, 20 μL of D-luciferin (150 mg/mL) (XenoLight, Perkin Elmer, Boston, MA, USA) were added to each well and plates were analyzed in the IVIS^®^ Lumina LT Series III equipment (Xenogen Corp., Alameda, CA, USA) using the following parameters: automatic exposure time, F/stop 4, binning of 8 and FOV of 12.9 cm with 3 min of the interval between each image acquisition, over a total of 297 min. The kinetics of bioluminescent signal were recorded at photons/s and analyzed using Living Image Software v.4.7.3 (IVIS Imaging System, Boston, MA, USA).

#### 2.2.5. Multimodal Superparamagnetic Iron Oxide Nanoparticles

BM-MNC were labeled with Superparamagnetic Iron Oxide Nanoparticles (SPION) with magnetic and fluorescence properties that are composed by a crystalline iron oxide (Fe_3_O_4_) nucleus of 8 nm coated with dextran, an average hydrodynamic size of 35 nm, conjugated with two types of fluorophores, one that emits NIRF excitation/emission wavelengths in the 750/777 nm range and another the Rhrodamina-B (Rh) in the 558/580 nm range, and shows a density of 1.25 g/cm^3^ and 6.4 × 10^16^ SPION_NIRF-Rh/_g (Biopal, Molday ION ™, Worcester, MA, USA).

#### 2.2.6. Polydispersion, Stability, Optical Caracterization and Zeta Potention Analysis of SPION_NIRF-Rh_

The polydispersion evaluation of SPION_NIRF-Rh_ hydrodynamic size suspension in water was performed with 30 µg Fe/mL of concentration using the dynamic light scattering (DLS) technique with the Zetasizer Ultra system (Malvern, Worcestershire, UK), where the polydispersion curve of hydrodynamic size was obtained at an angle of 173°, with the number of averages set at 30 and acquisition time of 3 s, in a fixed position at 37 °C, with a 120 s of thermic equilibrium. The mean diameter and standard deviation were obtained by adjusting the experimental data to a log-normal distribution function.

The stability of hydrodynamic diameter over time was evaluated with SPION_NIRF-Rh_ dispersed in (i) water (control); (ii) SFEM; and (iii) SFEM+FBS, in 50 µg Fe/mL of concentration. The recording of the size polydispersion curves was acquired every 30 min for 420 min at 37 °C.

The optical characterization of SPION_NIRF-Rh_ was performed at 30 µg Fe/mL dispersed in aqueous medium contained in a quartz cuvette of 1 cm of optical path using an RF-6000 spectrofluorophotometer (Shimadzu, Kyoto, Japan) and the LabSolutions RF software (Shimadzu, Kyoto, Japan). The 2D and 3D excitation spectrum graphics were acquired in the wavelength range from 400 to 850 nm and emission from 450 to 870 nm.

The zeta potential evaluation also was performed with the Zetasizer Ultra system using 100 µg Fe/mL of SPION_NIRF-Rh_ in water, pH 7.4 at 37 °C.

#### 2.2.7. BM-MNC Labeling with SPION_NIRF-Rh_

For BM-MNC labeled with SPION_NIRF-Rh_, cells of young and old mice isolated were plated at 1 × 10^6^ cell/mL in a 24-well plate with complete SFEM. Then, SPION_NIRF-Rh_ was added to each well at a concentration of 50 μg/mL for 4 h at 37 °C. After this period, the cells were washed 3 times with PBS, centrifuged at 500× *g* for 5 min at 21 °C to remove the SPION_NIRF-Rh_ which was not internalized into the cells.

#### 2.2.8. Internalization of SPION_NIRF-Rh_ into BM-MNC

The analysis of SPION_NIRF-Rh_ internalization into BM-MNC was performed by brightfield and fluorescence microscopy due to the magnetic (iron oxide) and optical (Rhodamine-B) properties of SPION_NIRF-Rh_. Ten microliter suspensions of BM-MNC of young and old mice in PBS were deposited on microscope slides for these analyses. Brightfield microscopy analysis was performed with the Prussian blue staining that highlights the nanoparticles, staining the iron presence in blue color. Therefore, 10 μL of this staining solution containing 5% potassium ferrocyanide (Sigma Aldrich, St Louis, MO, USA) and 5% hydrochloric acid (Merck, Darmstadt, Germany) was added on the slide containing the cells for 10 min and washed once with Milli-Q^®^ water (EMD Millipore Corporation, Bedford, MA, USA). Then, the Nuclear Fast Red staining was performed with a 1% solution (0.02 g of Nuclear Fast Red in 2 mL of deionized water) for 5 min for nuclear counterstaining, and the nuclei were quickly washed once more and analyzed by optical microscopy.

The fluorescence analysis was performed after BM-MNC nucleus labeling with 4′,6-diamidine-2′-phenylindole dihydrochloride (DAPI) for 5 min, followed by washing with water. The fluorescence images were acquired using an excitation/emission filter of 358⁄461 nm for DAPI and another of 530/550 nm for Rhodamine-B that is coupled in SPION_NIRF-Rh._ The images were obtained under Nikon TiE microscopy (Nikon, Tokyo, Japan).

#### 2.2.9. BM-MNC Viability after Labeling with SPION_NIRF-Rh_

BM-MNC viability after labeling with SPION_NIRF-Rh_ was analyzed through the bioluminescence technique. This analysis was performed using different concentrations of SPION_NIRF-Rh_ 10, 30 e 50 μg Fe/mL to the BM-MNC labeling of young and old mice. After 4 h of labeling, the cells were washed and transferred to Eppendorf tubes, to which was added 20 μL of luciferin in each Eppendorf, and then the BLI images were acquired using the IVIS^®^ Lumina LT Series III with automatic exposure time, F/stop 4, binning of 8 and FOV of 12.9 cm. For BLI intensity analysis we selected a region of interest of 2.5 cm^2^. The signals acquired were analyzed using Living Image Software version 4.7 in units of photons/s and the cellular viability was obtained by the relation (BLI intensity of BM-MNC labeled/BLI intensity of BM-MNC unlabeled) × 100.

#### 2.2.10. Signal and Quantification Analysis of the SPION_NIRF-Rh_ Loud Internalized into BM-MNC Using NIRF, ICP-MS, and MRI Techniques

The quantification analysis of SPION_NIRF-Rh_ internalized into BM-MNC of young and old mice after labeling was performed by NIRF, ICP-MS, and MRI techniques.

NIRF: The quantification of SPION_NIRF-Rh_ internalized into BM-MNC by NIRF was performed with cells labeled with 10, 30, and 50 μg/mL of SPION_NIRF-Rh_. Firstly, we made a calibration curve from the known SPION_NIRF-Rh_ concentrations of 2, 5, 10, 12, 16 µg Fe/mL and their respective fluorescence intensities (NIRF). For the curve adjustment we performed a linear regression (SPION_NIRF-Rh_ concentration vs. NIRF intensity). Then, the NIRF intensity of BM-MNC labeled was extrapolated using the calibration curve and we performed the SPION_NIRF-Rh_ quantification per BM-MNC. For NIRF images’ acquisition we used an excitation source of 710 nm and an emission wavelength of 780 nm using the IVIS Spectrum system (Xenogen, Xenogen Corp., Alameda, CA, USA).

ICP-MS: The quantification of iron content internalized into BM-MNC after SPION_NIRF-Rh_ labeling was performed by the mass spectroscopy technique using the Inductively Coupled Plasma Mass Spectrometry (ICP-MS) model Nexion 350× (Perkin Elmer, Boston, MA, USA). For this we used 1 × 10^6^ cell/mL labeled at the following SPION_NIRF-Rh_ concentrations: 10, 30 e 50 μgFe/mL dispersed in 100 µL of PBS and 300 µL of nitric acid (37%) for digestion over 4 h at 70 °C. The samples digested were diluted 200 times with Milli-Q^®^ water and analyzed by ICP-MS to determine the iron content of samples. The measures were performed in sextuplicate and the quantification was based on the calibration curve using the certified standard iron (NexION # N8145054).

MRI: After 4 h of BM-MNC labeled with 50 μg Fe/mL of SPION_NIRF-Rh_ the cellular samples were mixed and homogenized with 1% of agarose (Sigma Aldrich, St Louis, MO, USA) and transferred to a 96-well plate. The MRI images of BM-MNC labeled with SPION_NIRF-Rh_ dispersed in agarose were acquired using the 3T MRI equipment (Magnetom Vision, Siemens, Erlangen, Germany) for the entire body with a 32-channel head coil using a T2-weighted sequence (multicontrast turbo-spin echo, SE_MS); different echo times (TE = 8, 16, 24, …, 256 ms); fixed repetition time (TR = 3000 ms); slice thickness of 3 mm; FOV of 15.9 × 20.0 cm^2^; matrix of 256 × 256 and 16 averages. Then, the images were analyzed by SyngoVia software (Siemens, Erlangen, Germany) using a ROI for each well image to extract the MRI intensity signals acquired in the different TEs. From the adjustment of the exponential decay curves, I = I_o_exp(-TE/T2), where I is an intensity signal and Io is an initial intensity signal of resonance magnetic, were obtained the values of T2 transverse relaxivity time. To calculate the amount of SPION_NIRF-Rh_ internalized in the cells, it was used: 1/(T_2_)^BM-MNCS labeled^ = 1/(T_2_)^control^ + r_2_ × [SPION_NIRF-Rh_], with transverse relaxivity r_2_ of (19.9 ± 0.9) × 10^−4^ ms^−1^ µgFe^−1^ mL. Then, we calculated the number of SPION_NIRF-Rh_ per cell using the following relation: Number_SPION_NIRF-Rh_ = 6 × [SPION_NIRF-Rh_] × (Fe atomic mass) / (π × ρ_SPION__NIRF-Rh_ × M_Fe_ × ∅_SPION__NIRF-Rh_), where [SPION_NIRF-Rh_] is the iron load internalized in cells (gram); ρ_SPION__NIRF-Rh_ iron oxide density; M_Fe_ Fe molecular weight; and ×∅_SPION__NIRF-Rh_ SPION_NIRF-Rh_ diameter.

### 2.3. In Vivo Study

#### 2.3.1. BMT Model

For the BMT experiments, young 6-8-week-old C57BL/6 mice were used as recipients. Mice were irradiated with 9 Gy in a single fraction in an X-ray radiator Rad Source RS2000 Biological System (Buford, GA, USA) of the Cancer Institute of the State of Sao Paulo (ICESP), 24 h before the transplant. For the transplantation, recipient mice were anesthetized and injected with a suspension of 3 × 10^6^ labeled BM-MNC in 150 μL of PBS via retroorbital plexus. The injection was performed with a 27G needle with an infusion speed of 50 μL/min controlled by a micro drop infusion pump (Pump 11 Elite Nanomite, Holliston, MA, USA).

#### 2.3.2. BM-MNC Labeled with SPION_NIRF-Rh_ Homing Evaluation by NIRF

The NIRF homing evaluation of BM-MNC labeled with SPION_NIRF-Rh_ was performed 2, 4, 24 and 72 h after cell transplantation. The NIRF images were acquired using the IVIS Spectrum system equipment in the dorsal animal position with the following parameters: exposure time of 3 s, F/stop 4, binning 8, FOV of 12.9 cm, excitation filter of 710 nm, and emission of 780 nm. At the end of the experiments, the animals were euthanized to obtain ex vivo images of selected organs.

#### 2.3.3. BM-MNC Labeled with SPION_NIRF-Rh_ Tracking and Their Grafting by BLI

Kinetics of HSC engraftment was monitored by BLI over 4 months after BM-MNC transplantation. During the first month, the BLI images were acquired weekly, and for the following months the images were taken approximately every 15 days. Animals received 150 mg/kg of D-luciferin (i.p.) 10 min before BLI. The images were acquired on the IVIS Lumina equipment in the dorsal and ventral positions, using an automatic exposure time, F/stop 4, binning 8 and FOV 12.9 cm. For the acquisition of images, the animals were trichotomized before the acquisition to reduce the absorption of light by the melanin of the fur.

#### 2.3.4. Histological Analysis of Iron Present in Tissues

We euthanized 4 mice 24 h after BMT and some organs (brain, lung, heart, liver, kidney, spleen, and bone marrow) were collected, fixed in 4% paraformaldehyde for 24 h. Bones were decalcified using an aqueous solution of ethylenedinitrilo tetraacetic acid-dehydrated disodium salt (EDTA). After fixation, the tissues were dehydrated using ascending concentrations of alcohol (70%, 90%, 4× 100%), clarified in three batches of xylol solutions, and immersed in paraffin (two 1-h immersion batches). Then, the tissues were cut to a thickness of 5 μm in a microtome (Leica, Buffalo Grove, IL, USA) and stained with Prussian blue and Nuclear fast red. The slides were viewed using a Nikon TiE fluorescence microscope.

#### 2.3.5. Evaluation of Hematological Reconstitution (Blood Count and LKS Quantification)

The hematological reconstitution was evaluated after 15, 30, 60, 90, and 120 days of transplantation by blood cell count using peripheral blood samples collected with EDTA. The quantitative evaluation of red blood cells, leukocytes, and platelets was performed using the hematological analyzer Hematoclin 2.8 Vet (Bioclin, Belo Horizonte, Minas Gerais, Brazil).

The quantification of LKS cells present in BM recipients was performed by flow cytometry analysis after 120 days of BMT, using the BD Mouse Hematopoietic Stem and Progenitor Cell Isolation Kit, as previously described above (item 2.2.2.), and the data obtained were analyzed using FACSDIVA and FlowJo software.

### 2.4. Statistical Analysis

Data were presented as the mean and standard deviation in each analysis. For the in vitro study, the cellular viability and iron load quantification of the SPION_NIRF-Rh_ concentrations of young and old BM-MNC were compared by the ANOVA test, following post hoc tests corrected by Bonferroni or Student’s *t*-test for two samples. Previously, we analyzed the normal distribution and homoscedasticity of the data of each group, and we considered the significative level *p* < 0.05. All statistical analysis was performed with the JASP software v.0.14.1 (http://www.jasp-stats.org; Accessed on 1 January 2020).

## 3. Results

### 3.1. In Vitro Study

#### 3.1.1. Immunofenotypic Characteristics of Bone Marrow Mononuclear Cells (BM-MNC)

The characterization of the LSK cells from young and old mice was performed with and without previous administrations of 5-FU by flow cytometry analysis (Figure 1). We observed an increase in the population of lineage- cells with the 5-FU treatment, from 18.9% to 59.5% in young (Figure 1A,B) and from 13.4% to 91.6% in old cells (Figure 1E,F), indicating a notable enrichment of hematopoietic progenitors. Young mice displayed higher frequencies of lineage+ cells in BM after 5-FU treatment, perhaps due to a superior ability of HSCs to return to homeostasis when compared to old mice.

Regarding the hematopoietic stem/progenitor compartment, LKS in young animals’ BM represented 5.13% of the 18.90% of lineage-cells (0.97% of total cells) in untreated animals (Figure 1A,C, respectively), and 4.27% of the 59.50% of lineage-cells (2.54% of total cells) in 5-FU-treated mice (Figure 1B,D, respectively). However, the LKS in old animals’ BM represented 0.13% of 13.40% of lineage-cells (0.02% of total cells) in untreated animals (Figure 1E,G, respectively), and 0.93% of 91.65% of lineage-cells (0.85% of total cells) in mice treated with 5-FU (Figure 1F,H, respectively). Thus, 5-FU treatment led to a 3-fold LKS cell enrichment in young mice as opposed to 49-fold enrichment in old animals. However, the total frequency of LKS in bone marrow after 5-FU treatment was still three times higher in young mice as compared to their old counterparts (2.54% versus 0.85%).

#### 3.1.2. Time Course of Activity of Luciferase between Young and Old Bone Marrow Mononuclear Cells

The kinetics of luciferase activity were evaluated by in vitro BLI of young and old BM-MNC during 297 min using the different numbers of cells (1 × 10^4^, 2 × 10^4^, 5 × 10^4^, 1 × 10^5^, 2 × 10^5^).

Figure 2A shows the bioluminescent signal intensity of young BM-MNC in different concentrations. The peak of signal (9.3 × 10^9^ photons/s) was detected in the wells with 2 × 10^5^ cells 21 min after the addition of D-luciferin, followed by a gradual decrease over time. This kinetics were similar in BM-MNC obtained from old mice (Figure 2B), with a peak signal of 8.6 × 10^9^ photons/s in the 2 × 10^5^ cell concentration at 21 min and a reduction in intensity signal over time. The signal decay can be visualized in the other BM-MNC concentrations evaluated: 1 × 10^5^, 5 × 10^4^, 2 × 10^4^ and 1 × 10^4^ after their respective peak signal values: 5 × 10^9^, 3.3 × 10^9^, 1.3 × 10^9^, 7 × 10^8^ photons/s for young cells, and 5 × 10^9^, 3 × 10^9^, 1.4 × 10^9^ and 5 × 10^8^ photons/s for old cells.

In addition, we performed exponential fit curves of the peak signal intensity of each cell concentration analyzed to evaluate the behavior of bioluminescence kinetics between the young and old murine cells, in which we observed similar exponential behavior of intensity signal after adjustment, as shown in Figure 2C.

#### 3.1.3. Analysis of SPION_NIRF-Rh_ Polydispersion, Stability, Optical Characterization, and Zeta Potential

The polydispersion size analysis of SPION_NIRF-Rh_ in water suspension showed that after adjusting to a log-normal distribution, the average hydrodynamic diameter was 38.2 ± 0.5 nm, as shown in Figure 3A.

This analysis was also performed in the following suspension conditions: SFEM; H_2_O + SPION_NIRF-Rh_; SFEM + SPION_NIRF-Rh_; FBS + SPION_NIRF-Rh_; SFEM + FBS + SPION_NIRF-Rh_, as shown in Figure 3B. The SFEM sample showed a hydrodynamic diameter peak at 9.8 nm (green dashed curve of Figure 3B), the H_2_O + SPION_NIRF-Rh_ sample showed a peak at 37.3 nm (black curve of Figure 3B), but when the SPION_NIRF-Rh_ were suspended in SFEM we observed two peaks, one at 9.5 nm and other at 43.6 nm (red curve of Figure 3B): according to previous results, the first peak represents the SFEM (9.5 nm) and the second the SPION_NIRF-Rh_ (43.6 nm), but the SPION_NIRF-Rh_ had a light agglomeration process increasing the diameter from 37.3 to 43.6 nm, a change that did not interfere in the internalization process due to low variability. In the FBS + SPION_NIRF-Rh_ condition the peak was observed at 32.4 nm, which represents the SPION_NIRF-Rh_ diameter without evidence of FBS influence in SPION_NIRF-Rh_ stability. The last condition (SFEM + FBS + SPION_NIRF-Rh_) represents one possibility of cell labeling that is FBS associated to SFEM, during which we observed two peaks at 8.3 nm and at 42.7 nm (blue dashed curve in Figure 3B), similar to the results of SFEM + SPION_NIRF-Rh_ condition, confirming again that the FBS does not interfere in the polydispersion of the SPION_NIRF-Rh_. The inset in Figure 3 shows the three polydispersity spectra that allow the understanding of the solute in the cellular labeling process, in which SPION_NIRF-Rh_ are not affected by the presence of SFEM or FBS.

The hydrodynamic diameter stability of SPION_NIRF-Rh_ was also evaluated temporally over 420 min in three of the conditions analyzed previously (H_2_O + SPION_NIRF-Rh_; SFEM + SPION_NIRF-Rh_; SFEM + FBS + SPION_NIRF-Rh_) from the polydispersion curves of the SPION_NIRF-Rh_ as shown in Figure 3C,E,G, respectively, for which the mean hydrodynamic diameter (maximum peak of the curve referring to SPION_NIRF-Rh_) was obtained over time as shown in the graphs of Figure 3D,F,H. The conditions represented in Figure 3E,F correspond to the interaction of SPION_NIRF-Rh_ with the SFEM medium used in the in vitro and in vivo experiments, in relation to the control (H_2_O + SPION_NIRF-Rh_) represented in Figure 3C,D.

The optical characterization of SPION_NIRF-Rh_ shown in Figure 4A,B, highlights their spectrums of excitation (blue and green curves) and emission (red and orange curves). In Figure 4A, the spectrum shows the visible fluorescence peaks (562.2/588.3 nm, the blue and red curves, respectively) and the near-infrared fluorescence peaks (757.8/779.4 nm, the green and orange curves, respectively). In Figure 4B, the 3D visualization of SPION_NIRF-Rh_ excitation/emission with double fluorescence.

The zeta potential characterization analyzed by the surface charge of SPION_NIRF-Rh_ measured in H_2_O with a pH of 7.4, the same condition used in cell labeling with SPION_NIRF-Rh_, found a zeta potential of approximately 34.9 mV (Figure 4C). This positive potential provides a greater electrostatic interaction between the cells and SPION_NIRF-Rh_, favoring their internalization; the cells already have negative zeta potential.

#### 3.1.4. A SPION_NIRF-Rh_ Internalization into BM-MNC Evaluation by Brightfield and Fluorescence Microscopy

After the process of labeling the young and old BM-MNC with SPION_NIRF-Rh_, the internalization of SPION_NIRF-Rh_ was analyzed by brightfield and fluorescence microscopy (Figure 5A,B). The brightfield microscopy allowed us to visualize the BM-MNC, highlighting the intracellular iron of SPION_NIRF-Rh_ internalized in the cytoplasm of cells through Prussian blue staining. Fluorescence (Rhodamine-B) microscopy allowed us to visualize the presence of rhodamine fluorophore coupled with a SPION_NIRF-Rh_ through red imaging and the nucleus of BM-MNC labeled by DAPI staining, highlighted in the color blue.

Figure 5C shows the cell viability evaluation of young and old BM-MNC after labeling by BLI intensity images, in which we observed that the cells labeled with different SPION_NIRF-Rh_ concentrations (10, 30, and 50 µg/mL) retained similar viability, in comparison to the BLI intensity signal of unlabeled cells (control), independently of young or old BM-MNC (Table 1). The BLI signal intensity of young and old BM-MNC labeled with different SPION_NIRF-Rh_ concentrations and unlabeled (Figure 5D) were analyzed by ANOVA test, showing a significant difference (*p* < 0.001; Appendix A), but by post hoc analysis the significant difference did not occur between young and old BM-MNC (*p* = 1) for any labeled concentrations; it was only found between concentrations of labeling when compared the control (BM-MNC unlabeled) with concentrations of 30 and 50 µg/mL (*p* < 0.001), independently of young or old BM-MNC (Appendix A). In addition, the bar graph of Figure 5E shows that the viability reduction was only about 5% in comparison to the higher concentration (50 µg/mL) of the control (unlabeled) which means a low reduction in cell viability even at the highest SPION_NIRF-Rh_ concentration analyzed.

#### 3.1.5. Quantification of SPION_NIRF-Rh_ Loud Internalized into BM-MNC by NIRF, ICP-MS, and MRI

NIRF: After the BM-MNC labeling with SPION_NIRF-Rh_, the NIRF signal intensity was analyzed, showing an increased NIRF signal intensity according to the increase in the concentration of SPION_NIRF-Rh_, both in young and old cells (Figure 6). From the NIRF calibration curve we obtained the following relation [NIRF intensity] = (7.23 ± 5.56) × 10^7^ + (1.91 ± 0.65) × 10^8^ × [SPION_NIRF-Rh_] and the iron load values (pg Fe/cell and SPION_NIRF-Rh_ number/cell) of the different SPION_NIRF-Rh_ concentrations internalized into BM-MNC. The values obtained ranged between 2 pg Fe/cell (10 µg Fe/mL of concentration) for young cells and for old cells until about 4 pg Fe/cell (50 µg Fe/mL of concentration), as shown in Table 2 and Figure 6—NIRF Quantification (blue and red boxplots correspond to the young and old BM-MNC, respectively). The analysis of these values by ANOVA test showed a significant difference between the iron load values (*p* < 0.001; Appendix A), but the post hoc analysis verified that the difference did not occur between young and old cells (*p* = 1) for any SPION_NIRF-Rh_ concentrations (10, 30, and 50 μg Fe/mL) analyzed, with significant differences in iron load only occurring between concentrations of 10 to 30 or 50 μg Fe/mL of SPION_NIRF-Rh_ (*p* < 0.001) for both BM-MNC (Appendix A).

ICP-MS: The iron load quantification by ICP-MS was obtained from a calibration curve that related the ICP-MS intensity signal expressed in counts per second (cps) as a function of the iron concentration (ppb) and the equation of the linear adjustment of the calibration curve [Fe] = (−47.8 ± 0.72) + (5.27 ± 0.04) × 10^4^ × [Intensity]. The values obtained ranged from 1.90 ± 0.10 pg Fe/cell (10 µg Fe/mL of concentration) for young cells and 1.67 ± 0.06 pg Fe/cell for old cells to 4.23 ± 0.09 pg Fe/cell (50 µg Fe/mL of concentration) for young cells and 4.00 ± 0.07 pg Fe/cell for old cells, as shown in Table 2 and Figure 6—ICP-MS Quantification (blue and red boxplots correspond to the young and old BM-MNC, respectively). These iron load values (pg Fe/cell and SPION_NIRF-Rh_ number/cell) were analyzed by ANOVA test that showed a significant difference between values (*p* < 0.001, Appendix A), and the post hoc analysis showed that the difference in iron load occurred between young and old BM-MNC for the same SPION concentration analyzed (*p* < 0.01, Appendix A), as well as between SPION concentrations (10, 30, and 50 µg Fe/mL) used in young or old BM-MNC labeling (*p* < 0.001, Appendix A).

MRI: For the iron load analysis by MRI quantification we used the phantom containing young and old BM-MNC labeled with 50 µg Fe/mL of SPION_NIRF-Rh_ and unlabeled, in which we acquired the MRI intensity as a function of the different echo times (image of Figure 6). In the graphic, the transverse relaxation curves of the signal intensity of the MRI as a function of each time allowed us to quantify the T2 values shown by the boxplot (blue and red colors correspond to young and old BM-MNC, respectively), and these values were compared by ANOVA test, which showed significant difference (*p* < 0.001, Appendix A), but through the post hoc analysis it was observed that the significant difference of iron load did not occur between young and old BM-MNC (*p* = 1; Appendix A), but occurred between the labeled and unlabeled conditions (*p* < 0.001; Appendix A).

For SPION_NIRF-Rh_ concentration of 50 µg Fe/mL, the only concentration performed by this technique, the iron load was 3.13 ± 0.24 pg Fe/cell or (3.93 ± 0.30) × 10^4^ SPION_NIRF-Rh_/cell for young BM-MNC and 3.08 ± 0.17 pg Fe/cell or (3.87 ± 0.21) × 10^4^ SPION_NIRF-Rh_/cell for old BM-MNC, as shown in Table 2 and Figure 6—MRI Quantification. These iron loads did not differ significantly between young and old BM-MNC by Student’s *t* test or in any of the measures analyzed (pg Fe/cell: *p* = 0.764; SPION_NIRF-Rh_ number/cell: *p* = 0.770) using Student’s *t* test (Appendix A).

### 3.2. In Vivo Study

#### 3.2.1. BMT of Young to Young and Old to Young Mice

BM-MNC Labeled with SPION_NIRF-Rh_ Homing and Tracking by NIRF: The BM-MNC homing of young and old donors labeled with SPION_NIRF-Rh_ was evaluated in vivo by NIRF for 72 h after BMT (Figure 7). The NIRF images were acquired in the dorsal plane, in which we observed the distribution of the labeled cells throughout the whole animal, with higher signal intensity in the spinal cord and abdominal region, possibly due to the presence of the liver and gastroenteric system (Figure 7). In addition, the NIRF signal in the tibiofemoral region in the recipient animals was maintained throughout the analysis in the group that received young BM-MNC but disappeared within 72 h in the group that received old BM-MNC. To refine the NIRF signal localization, we acquired ex vivo images of the main organs at 24 and 72 h after BMT (Figure 8) and detected NIRF signals in all hematopoietic organs (spleen, spine, tibia and femurs) as well as in the liver and gastroenteric system of mice transplanted with both young and old BM-MNCs. However, it was not possible to evidence differences in the NIRF signal between animals that received cells from young and old donors until 24 h, indicating that the homing of BM-MNCs was similar between young and aged donors.

BM-MNC Labeled with SPION_NIRF-Rh_ Tracking and Grafting Evaluation by BLI: The BLI of young mice that received young cells or old cells for 120 days (Figure 9) showed that on the fourth day after transplantation it was not possible to detect any bioluminescent signal. However, a signal was successfully detected from the 11th day on the dorsal plane, having a higher intensity in the left side of the animals, possibly due to the presence of the spleen, and also in the spinal cord and the tibiafemoral region, which is consistent with hematopoietic colonization. Over the course of the temporal evaluation, the bioluminescent signal became more intense, achieving a peak of maximum intensity 30 days after transplantation for the recipients of old cells (7.88 ± 2.61 × 10^8^ photons/s) and 35 days for the recipients of young cells (10.18 ± 2.35 × 10^8^ photons/s), as shown in the images and graphic of Figure 9 and Table 3. After its maximum intensity peak, the BLI signal greatly decayed until day 85, with respective BLI intensity values of 2.10 ± 1.19 × 10^8^ and 0.96 ± 0.21 × 10^8^ photons/s for the animals in receipt of young and old cells, which remained stable at approximately 10^8^ photons/s until the end of the analysis at 120 days (Table 3), indicating the presence of donor cells from both young and old animals throughout the evaluation period.

Figure 10 shows the in vivo and ex vivo BLI signal evaluation of animals and their main organs (brain, lung, heart, spleen, liver, kidneys, femur, tibia, and spinal cord) after 120 days, in which the patterns of BLI signal was similar between animals and the organs evaluated that receive young cells (Figure 10A–C), or old cells (Figure 10D–F). However, there was a prominent BLI signal in the spleen, lung, tibia, and spinal cord among the evaluated organs, which correlates with the high BLI signal of in vivo evaluation.

#### 3.2.2. The Iron Presence in Tissues Evaluated by Histological

The histological images of the main organs (kidney, spleen, brain, heart, liver, intestine, lung, and spinal cord) showed the presence of iron in almost all organs, with the exception of the brain (Figure 11). The presence of iron in tissues was highlighted by the blue color staining of SPION_NIRF-Rh_, being the most prevalent color in hematopoietic organs, such as the spleen and spinal cord.

#### 3.2.3. Evaluation of Hematological Reconstitution (Blood Count and LKS Quantification)

In the first analysis, hematological evaluation by blood count for 120 days showed similar values for all blood components in the young–young group and the old–young group, but inferior to the control, except granulocytes, which had values close to the control (Figure 12). The pattern of hematological reconstitution of the old–young group was slower than the young–young group over 120 days. Leukocytes and lymphocytes of the young–young and old–young groups were reestablished, achieving values close to control at the end of the analysis (Figure 12A,D). Monocytes of both experimental groups had similar behavior but maintained values inferior to control over 120 days (Figure 12C); however, granulocytes of both experimental groups obtained values above control over 120 days, and after 60 days the old–young group achieved values higher than the young–young group (Figure 12B). Regarding the values of red blood cells and platelets of both experimental groups, these were reestablished at 30 days, and remained close to control until 120 days (Figure 12E,F).

The quantification of LKS cells of recipient BM by flow cytometric analysis was performed 120 days after BMT, as shown in Figure 13, in which it is possible to observe that we did not obtain a difference between the cell population of CD117+ lineage- inside BM-MNC collected of the receptor of young (6.1%) and old cells (4.7%), as shown in the first graphic of Figure 12. Additionally, no difference occurred in the percentage of LKS cells inside of the CD117+ lineage- of the receptor of young (67.0%), and old cells (70.4%), as shown in the second graphic of Figure 13.

## 4. Discussion

Bone marrow transplantation is a widely used treatment for pathologies of the hematopoietic system; however, there are many challenges facing the improvement of the comprehension of HSC migration after transplantation, and one of these involves the lack of consistent data on HSC homing to BM, and also the survival of cells after their administration, and the grafting process of the varied HSC populations in in vivo experiments, using non-invasive techniques to allow the temporal evaluation in unique animals and in real-time.

Aging has a great influence on these questions; it is already known that HSC from old animals has a lower capacity for migration and grafting in comparison to HSC from young animals [21,22]. To better understand HSC homing after BMT and evaluate the influence of aging on cell migration and engraftment, we labeled BM cells from young and old mice with SPION_NIRF-Rh_ that contain the nucleus of iron oxide and two types of fluorophores in their surfaces with different wavelengths (visible and infrared spectrums): further, these cells were transduced with a lentiviral vector to express the luciferase, and these combinations of techniques allowed us to perform quantitative and qualitative analysis in vitro and in vivo. Therefore, the tracking agent choice, cell isolation care, and their labeling before the transplantation were challenges studied to potentialize the best conditions to achieve the objective of HSC homing and grafting evaluation.

Regarding the nanoparticles, for their use as tracking agent, it is necessary that they have adequate physical chemistry properties and that they contribute to an efficient internalization, in addition to maintaining good cell viability and the ability of HSC to engraft, thus allowing non-invasive cell tracking by molecular imaging techniques [23].

The SPION_NIRF-Rh_ were evaluated as their physical–chemical proprieties obtained a hydrodynamic diameter of 38.2 nm with a positive zeta potential of +34.9 mV; this positive value eased the internalization process of SPION_NIRF-Rh_ into BM-MNC since the cells had a negative load in their surface, contributing in the favor of the electrostatic process involved [24]. There are other strategies to potentialize the internalization process of nanoparticles into HSC, such as the use of transfection agents including polybrene, protamine sulfate, and lipofectamine [25,26,27].

The analysis of the hydrodynamic size of SPION_NIRF-Rh_ suspended in different solutions used for application showed variation according to the solution used to suspend the nanoparticles, due to the linkage of nanoparticles to proteins and other electrolytes dispersed in the solutions, leading to the formation of a “crown” and the increase in the nanoparticle size and/or appearance of other populations from the elements that contained the suspension medium [28,29]. Our study showed a 14% increase in diameter of SPION_NIRF-Rh_ dispersed in the culture medium (SFEM) used in the cells’ labeling, in addition to showing another peak probably that corresponded to some of the proteins present in SFEM, with a hydrodynamic diameter of 9.3 nm. According to a study by Sousa de Almeida [30] in this condition is possible to affirm that the physical chemistry characteristics of SPION_NIRF-Rh_ are favorable to the process of internalization into cells [30,31], as well as the probable endocytic pathway of internalization [30,31,32,33]. However, when this same condition of SPION_NIRF-Rh_ +SFEM was supplemented with 10% FBS it was possible to observe little variability. Generally, four forces are present in the interaction of magnetic nanoparticles with the culture medium (electrostatic forces, van der Waals forces, steric forces, and magnetic forces), and depending on the type of nanoparticle and the suspension medium there will be a competition between these forces so as to avoid the agglomeration of these nanoparticles [34,35].

Once we had measured the hydrodynamic size of the SPION_NIRF-Rh_ hydrodynamic size, we evaluated their stability over 4 h, which is the same time used during cell labeling with SPION_NIRF-Rh_ in other studies [25,26,27,36]. This evaluation showed stability of hydrodynamic size over time and in all conditions analyzed (Figure 3), even with the awareness that the hydrodynamic size of SPION_NIRF-Rh_ increase could affect the capacity of their internalization by the formation of nanoparticle clusters due to a loss of surface functionality, adsorption of proteins and electrolytes, resulting in the formation of aggregates, [30,34]; as a consequence of this force imbalance, SPION_NIRF-Rh_ is not kept in suspension, since depends on magnetic/electrostatic interaction, high energy of surface and van der Waals forces [34,35,37].

Another important aspect of performing the BMT experimental model is the limited number of HSC isolated from unique animals due to the low number of HSC present in BM, which requires a great number of mice to be used as donors [38]. Facing this challenge, we analyzed the effect of previous administrations of 5-FU in donor animals before cell isolation. This drug acts on the bone marrow inducing a temporary hypoplasia, which subsequently generates a stimulus for the BM to promote an increase in the production of HSC [39]. When we isolated the cells after 5-FU treatment we found a higher number of lineage- cells in both young and old mice, even though it was possible observed that LKS cells of BM from old mice were present in a low frequency in comparison to BM from young mice. This strategy of previous treatment with 5-FU potentialized cell tracking, whereas to evaluate the cell tracking in vivo in the non-invasive way it is necessary to administer a great number of cells [40,41]. In addition, we observed that after 4 days of treatment with 5-FU, the lineage- cell population from BM of young mice (2.54%) was of higher frequency than the BM of old mice (0.85%), due to the reduced differentiation capacity generated by aging, which ends up slowing the HSC response to 5-FU treatment [42].

The BM-MNC labeling/internalization from young and old mice with SPION_NIRF-Rh_ was performed only after SPION_NIRF-Rh_ characterization when their physical-chemistry features and stability in the in vivo application conditions. Generally, the verification of SPION_NIRF-Rh_ internalization into HSC as well as in other cells occurs by the Prussian blue staining, which evidences the nanoparticles internalized by the blue color [25,26,43,44]. However, in our study, beyond this analysis it was possible to corroborate the internalization of SPION_NIRF-Rh_ through fluorescence microscopy, MRI, and NIRF due to the magnetic and double fluorescence features of this nanoparticle. The evaluation of brightfield and fluorescence microscopies showed the SPION_NIRF-Rh_ in the region of the cell cytoplasm, with a high nucleus/cytoplasm ratio and in the range of 10 µm suggestive of HSC [45]. In NIRF evaluation we obtained a signal in the order of 10^8^ photons/s for all SPION_NIRF-Rh_ concentrations analyzed, and the signal was about 35% higher in the SPION_NIRF-Rh_ concentration of 50 than 10 µg/mL, in cells from both young and old mice (Figure 6). In addition, the cell viability after labeling with SPION_NIRF-Rh_ was evaluated by BLI in the same concentrations described above, and the cells labeled with the highest SPION_NIRF-Rh_ concentrations (50µg/mL) showed viability above 95% (Figure 5); however, another study that labeled HSC with nanoparticles reported viability below 90% in the highest nanoparticle concentration used in their labeling process (125 µg/mL) [27]. In vivo MRI analysis of BM-MNC from young and old mice labeled with 50 µg/mL of SPION_NIRF-Rh_ and their respective control (unlabeled) showed a hypointense signal in the labeled cells in comparison to control, and a 39% of reduction in the T2 transverse relaxation time in young in comparison to old cells (Figure 6).

The quantification of iron load internalized into BM-MNC by NIRF, MRI, and ICP-MS (Table 2) showed signal detection according to the signal detection limitation of each technique as a function of their sensibility, spatial, and temporal resolutions [17,46]. Some studies reported an internalization of 0.69 ± 0.08 [47]; 1.33 ± 0.2 [48]; 2.01 ± 0.10 pg [49] of nanoparticles per HSC. Our study found a higher internalization, as shown in Table 2, with values of 4.23 ± 0.09 pg Fe/BM-MNC or (5.32 ± 0.11) × 10^4^ SPION_NIRF-Rh/_BM-MNC from young mice, and in BM-MNC from old mice 4.00 ± 0.07 pg Fe/BM-MNC or (5.03 ± 0.81) × 10^4^ SPION_NIRF-Rh/_BM-MNC, by the ICP-MS technique. The quantification by MRI confirmed these high values of internalization 3.13 ± 0.24 pg Fe/BM-MNC or (3.93 ± 0.30) × 10^4^ SPION_NIRF-Rh/_BM-MNC from young mice, and in BM-MNC from old mice found 3.08 ± 0.17 pg Fe/BM-MNC or (3.87 ± 0.21) × 10^4^ SPION_NIRF-Rh/_BM-MNC. NIRF quantification also showed high values of iron load internalization, 3.98 ± 0.16 pg Fe/BM-MNC or (5.00 ± 0.20) × 10^4^ SPION_NIRF-Rh/_BM-MNC from young mice, and in BM-MNC from old mice 3.98 ± 0.14 pg Fe/BM-MNC or (5.00 ± 0.18) × 10^4^ SPION_NIRF-Rh/_BM-MNC. These values are higher than the cited studies, possibly because we used a cell pool of BM, beyond which we used small nanoparticles with a hydrodynamic size of 38.2 ± 0.5 nm, which is different to other studies, which reported nanoparticles with hydrodynamic sizes higher than 200 nm, which hinder internalization [47]. Additionally, it was possible estimate the number of SPION_NIRF-Rh_ internalized into BM-MNC, and all measures of the iron load by different techniques showed results in the order of 10^4^ SPION_NIRF-Rh_/BM-MNC.

The in vivo study sought to evaluate the homing and grafting of BM-MNC from young and old mice labeled with SPION_NIRF-Rh_ and modified to luciferase expression by BLI, NIRF and to corroborate histological results by Prussian blue staining. The acute evaluation of BM-MNC transplant by NIRF, after 24 h of cells administration, showed a signal detection for whole animal bodies similar to the results that were reported in the study by Ushiki [50]. After in vivo image evaluation, the animals were euthanized, and their organs were collected to perform the ex vivo evaluation, wherein the NIRF signal presence in main organs, including the related to hematopoiesis as BM and spleen, was evident: these results was corroborated by histological analysis that evidenced the blue color of the iron contained in SPION_NIRF-Rh_ in almost all tissues analyzed, with the exception of the brain and intestine, with accumulation in spleen and bone marrow (tibia), similar to the findings reported by Maxwell’s study [51]. Other studies that evaluated the acute HSC homing to BM by molecular image techniques (PET, NIRF, and MRI) also managed to visualize the transplanted cells in the recipient animal in the spleen and BM regions, and beyond this to other organs, indicating that homing is a fast process and that some of the cells go to other organs besides those that promote hematopoiesis [27,50,52,53].

The homing and tracking evaluation of transplanted cells to BM by BLI was effective only after the 11^th^ day of cell transplantation (out of the acute phase), but this technique allowed us to follow the cell grafting in the animals until the end of the analysis (120 days), whereby we obtained initial signal in the order of 2 × 10^8^ photons/s in receptors of young and old cells, with an increase over time until their peak intensity at 30 days (7.88 ± 2.61 × 10^8^ photons/s) for the receptor of old cells and 35 days (10.18 ± 2.35 × 10^8^ photons/s) for the receptors of young cells: from these peaks the BLI signal began to decay, stabilizing at 85 days, with a signal in the order of about 10^8^ photons/s until the end of the analysis at 120 days.

The pattern of BLI signal until 85 days was compatible with hematopoietic reconstitution analyzed by blood count, which verified a grafting response with a great number of mature cells in circulation at 30 days: this increase is probably a reflection of the graft trying to restore the function of the bone marrow, producing cells in large quantities [54]. Then, in the 60-day analysis, we obtained a reduction in the leukocyte values, similar to the reduction in the intensity of the BLI signal; however, in the following analyses, the number of leukocytes increased while the BLI signal decreased until the end of the analysis. These results may be de to the hematopoiesis reestablishment by HSC of the receptors that were not destroyed in the ICI process, and which are now participating again in the production of blood cells in recipient individuals [55]. At the same time, the BLI signal of our study was similar to that found in the study by Cao, [56] which obtained a curve with a similar tendency when transplanting HSC after a short time and observed a decrease in signal intensity from the 60th day until the end of the analysis (120 days). This association between HSC survival in the host after ICI, progenitors transplant, and long-term HSC may explain the recovery of nucleated cells towards normal values, but with BLI signal decay. Unfortunately, our study failed to differentiate the transplanted cell populations from those of the host, which would bring data that would enable a better understanding of this phenomenon [57].

The hemogram analysis showed a difference in the frequency of circulating leukocytes; young cell receptor mice had a greater number of granulocytes circulating in the first two hematological analyses, but from then on lymphocytes or leukocytes were the main cell type found, a typical pattern for mice [58]. While elderly cell recipients had a predominance of granulocytes from 60 days onwards, this skewed differentiation for the myeloid lineage is characteristic of the aging of HSCs, which also leads to a reduction in differentiation in the lymphoid lineage [12]. This result is compatible with the observation that the adaptive immune system (largely mediated by the lymphoid system) decreases immune efficacy in the elderly [59,60].

The BLI technique did not allow HSC visualization in the first days after BMT; however, it allowed us to follow the grafting from the 11th day to 120 days after transplantation. This technique normally allows a long time for evaluation and providing the follow up of the transplanted cells until their differentiation [56] (as was reported in the study by Steiner et al. [61] that evaluated the grafting over one year, and also in other studies that reported an evaluation periodicity by BLI similar to our study, [62,63,64,65] or even shorter periods, [57]), although the range depends on the objective of study. Some studies reported the tracking of cells by BLI since the first day after cell transplantation (acute phase) [57,62,64], whereas in others it was possible to observe the first BLI signal of cells in periods similar to our study [61,65,66]. This first BLI signal detection can be influenced from the number of cells transplanted to model used as a receptor [63,67]. Therefore, the concomitant evaluation using NIRF in the acute evaluation and BLI to follow up for a long time showed the importance of the complementary use of two or more techniques to evaluate the biological process in vivo in a non-invasive way [68]. Due to magnetic feature of SPION_NIRF-Rh_ could also have been carried out the MRI evaluation, [25,26,27] but as we did not have this equipment for animal scanner images, the evaluation was not carried out. Another possibility to potentialize the power of SPION_NIRF-Rh_ tracking would be the addition of a radiotracer in this nanoparticle, improving and bringing more data on the acute tracking of BM-MNC to bone marrow by the use of the nuclear image technique [69].

A number of studies performed the HSC tracking to BM in a non-invasive way using a range of molecular imaging techniques, such as BLI, [57,61,62,63,64,65,66] FLI, [50,70,71,72,73] MRI, [25,26,27] PET [52,74,75], and SPECT [76,77,78], but none of these studies used a multimodal approach with two or more techniques associated with HSC tracking evaluation in a noninvasive way to corroborate the results. The study by Sweeney et al. [27] used a nanoparticle functionalized with FITC and TRITC; however, because they are fluorescent agents that emit fluorescence in the visible spectrum, they end up competing with the autofluorescence of the tissues, making it impossible to carry out in vivo evaluations. Some studies performed the HSC tracking in a non-invasive way by use of fluorescence imaging in the visible spectrum with the use of animals that such as zebrafish that enable greater tissue penetration [70,72,73]. However, this evaluation is also limited to HSC homing and did not allow the graft analysis, due to loss of fluorophore signals after cell division. Lopes et al. [71] reported the HSC grafting analysis by fluorescence imaging in the visible spectrum using genetically modified animals (Axolotl White mutant (d/d)) that enabled greater tissue penetration and using transgenic donors that expressed GFP. Our study, on the other hand, used a nanoparticle that has two fluorophores, one that emits in the visible spectrum and the other in the NIRF, which makes it possible to trace the labeled cells in mice, with less interference from tissue autofluorescence and greater penetration capacity than fluorophores in the visible emission spectrum.

In summary, our results on in vivo HSC tracking in the bone marrow transplantation model did not show a difference in the migration or grafting of cells from young or old mice; only in the hemogram analysis did we observe a trend of the differentiation towards the myeloid lineage in mice that received cells from old animals. The combined use of SPION_NIRF-Rh_ and BLI is thus shown to be a promising and able candidate for in vivo cell tracking evaluation, allowing us to correlate the results to the quantitative data obtained via ex vivo analysis. However, for the grafting analysis, this strategy should be aligned with techniques that allow not only quantification but also evaluation of the quality of the grafting [52,72,74]. Nonetheless, our results demonstrate a limitation of the techniques used to evaluate HSC grafting in a more robust and long-term manner, which is extremely important information for the BMT model. Perhaps we could evidence differences in grafts in experiments with longer follow-up or using other cell types.

## Figures and Tables

**Figure 1 biomedicines-09-00752-f001:**
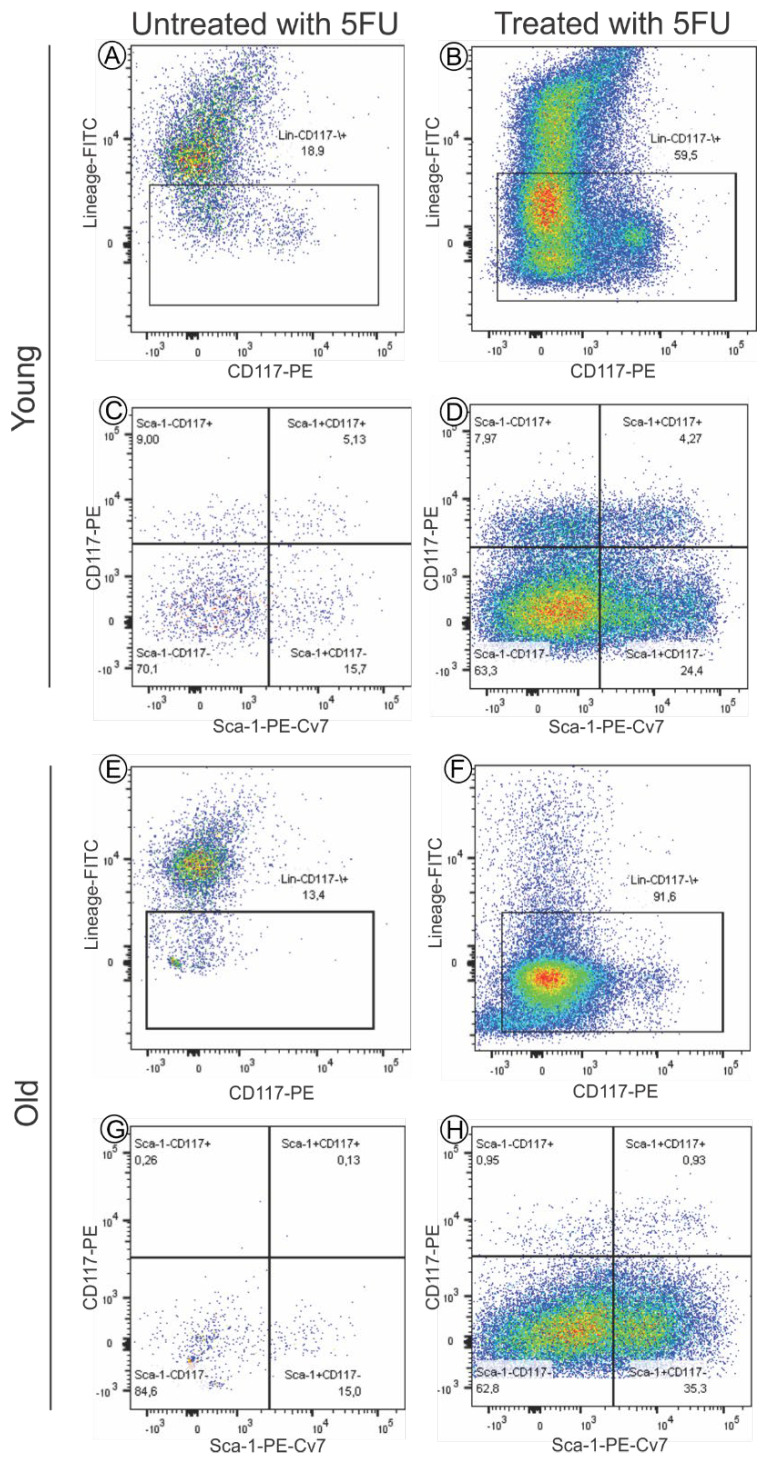
Flow cytometry comparative analysis of the population of LSK cells from young (**A**–**D**), and old (**E**–**H**) mice, with and without previous administration of 5-FU (treated and untreated with 5-FU), 4 days before cell isolation. (**A**,**B**,**E**,**F**) Analysis of CD117+ lineage- population within BM-MNC isolated of C57BL/6 mice; (**C**,**D**,**G**,**H**) Analysis of LKS population within cells of CD117+ lineage-.

**Figure 2 biomedicines-09-00752-f002:**
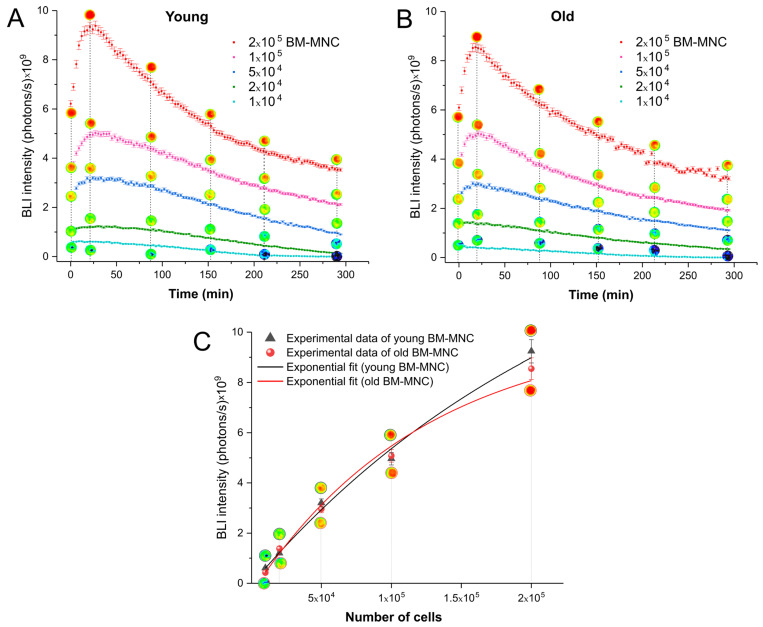
Bioluminescence signal intensity analysis of BM-MNC from (**A**) young and (**B**) old mice, as a function of the number of cells over 297 min. (**C**) Exponential fit curves of bioluminescence intensity peak of the different numbers of cells showing the Kinects between young and old mice BM-MNC after adjustment.

**Figure 3 biomedicines-09-00752-f003:**
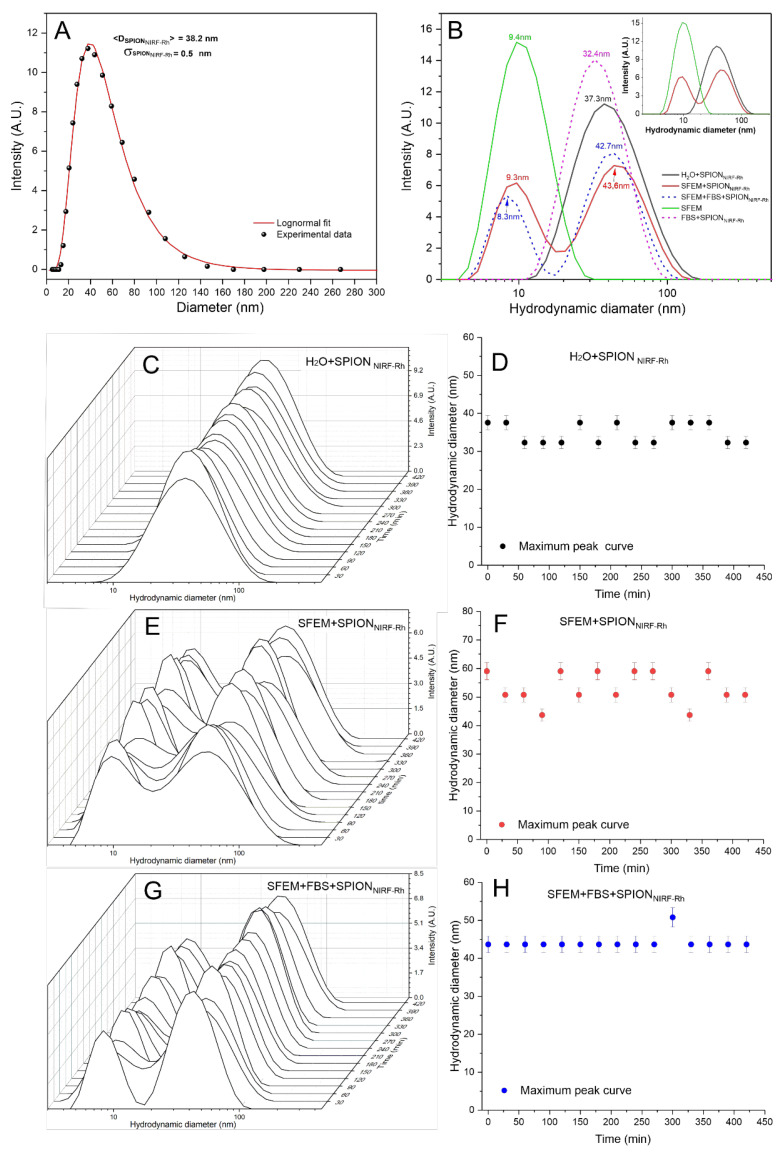
Polydispersion curve analysis of the hydrodynamic diameter of SPION_NIRF-Rh_ in water suspension (**A**); Polydispersion curve analysis of the hydrodynamic diameter of SPION_NIRF-Rh_ in different conditions of suspension (**B**); SPION_NIRF-Rh_ stability analysis in SFEM over 420 min (**C**–**H**); SPION_NIRF-Rh_ stability analysis in SFEM + FBS over 420 min (**E**,**F**); and SPION_NIRF-Rh_ stability analysis in H_2_O over 420 min (**G**,**H**).

**Figure 4 biomedicines-09-00752-f004:**
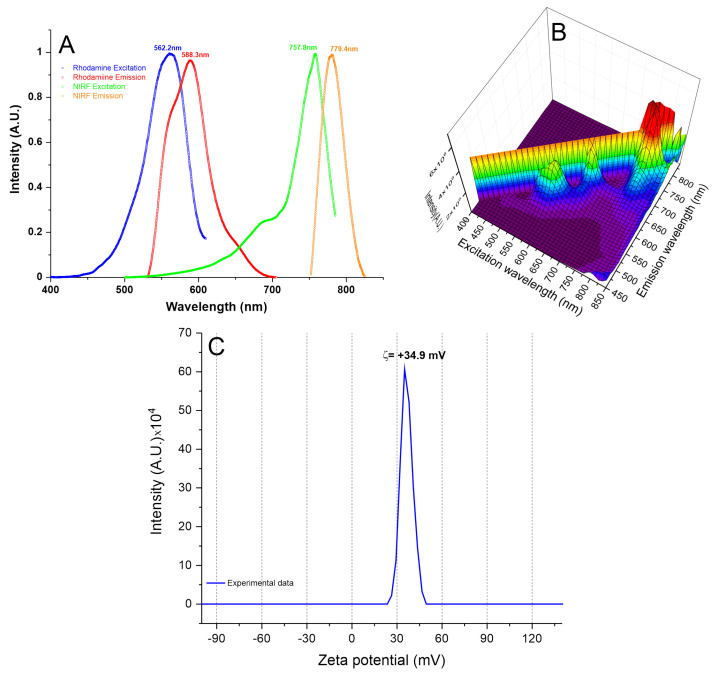
SPION_NIRF-Rh_ optical characterization. (**A**) the SPION_NIRF-Rh_ excitation/emission spectrum, pointing of the double fluorescence of this nanoparticle, with Rhodamine excitation and emission peaks at 562.2 and 588.3 nm (blue and red curves, respectively), and NP750 at 757.8 and 779.4 nm (green and orange curves, respectively). (**B**) the 3D SPION_NIRF-Rh_ spectrum, and (**C**) the zeta potential of SPION_NIRF-Rh_ with a surface charge at 34.9 mV.

**Figure 5 biomedicines-09-00752-f005:**
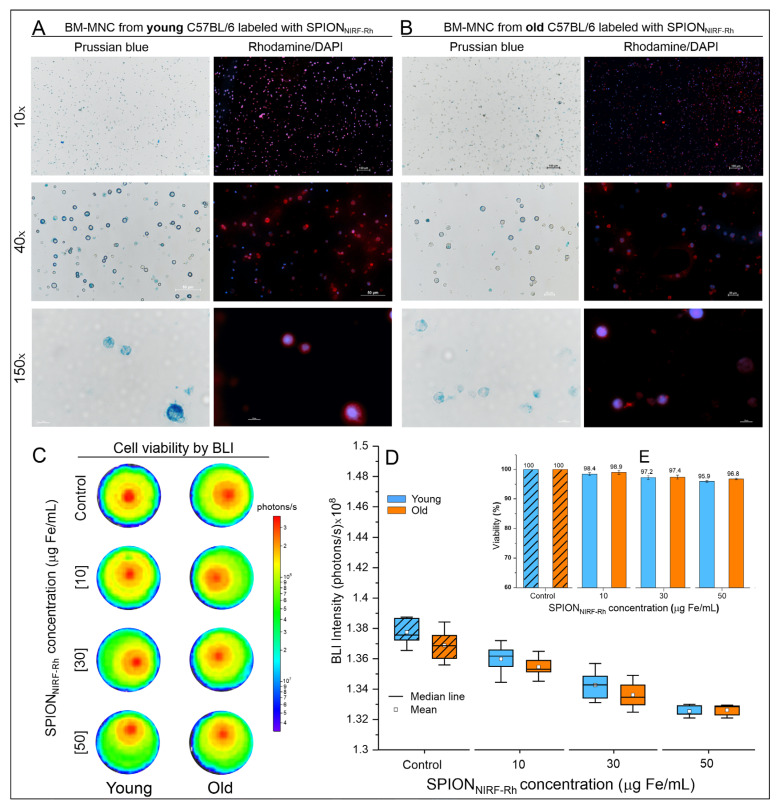
SPION_NIRF-Rh_ internalization analysis by brightfield and fluorescence microscopies and cell viability after labeling. (**A**) and (**B**) Brightfield and fluorescence microscopy of young and old BM-MNC labeled with 50 µg/mL of SPION_NIRF-Rh_. The images are shown in three amplifications (10×, 40×, and 100×) using the correspondent field of view in both microscopy analyses. (**C**) Cell viability analysis by BLI signal intensity of young and old BM-MNC labeled with different concentrations of SPION_NIRF-Rh_ (10, 30, and 50 μg Fe/mL) and the control (BM-MNC unlabeled). (**D**) Graphic representation of BLI signal intensity of young (orange boxplots) and old (blue boxplots) BM-MNC labeled with different concentrations of SPION_NIRF-Rh_ (10, 30, and 50 μg Fe/mL) and the control (BM-MNC unlabeled). (**E**) Bar graph of the BLI viability analysis of young (orange bars) and old (blue bars) BM-MNC after labeling with different concentrations of SPION_NIRF-Rh_ (10, 30, and 50 μg Fe/mL) and the control (BM-MNC unlabeled).

**Figure 6 biomedicines-09-00752-f006:**
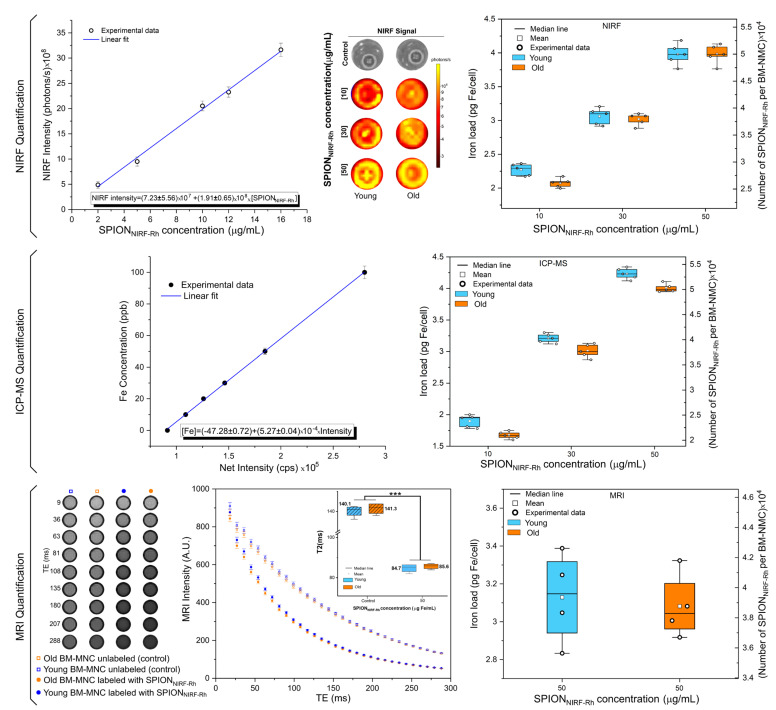
Quantification of SPION_NIRF-Rh_ load internalized into BM-MNC by NIRF, ICP-MS, and MRI. NIRF quantification shows the calibration curve used to calculate the NIRF intensity signal of each concentration of SPION_NIRF-Rh_ used (10, 30, and 50 µg/mL) in the young and old BM-MNC labeling in the boxplot graphic. ICP-MS quantification was also used as a calibration curve to show the SPION_NIRF-Rh_ load in the same SPION_NIRF-Rh_ concentrations already analyzed by NIRF. For MRI quantification we used the phantom containing young and old BM-MNC labeled with 50 µg Fe/mL of SPION_NIRF-Rh_ and unlabeled, and the T2 values were quantified through the transverse relaxation curves of the signal intensity MRI as a function of each time. The graphic comparison between groups in the same concentration was performed in the same way as above. Blue and red boxplots correspond to young and old BM-MNC, respectively. *** *p* < 0.001 as the comparison between the labeled and unlabeled conditions.

**Figure 7 biomedicines-09-00752-f007:**
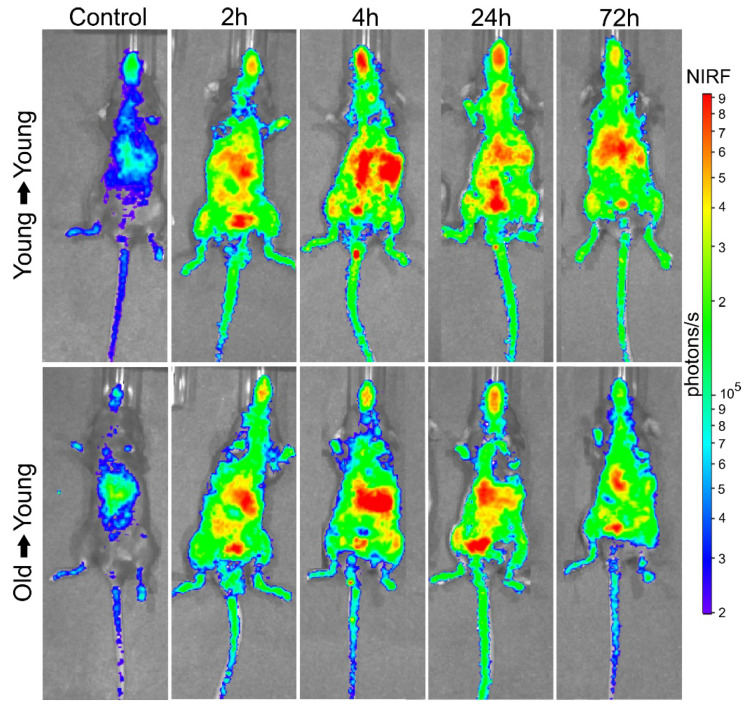
BM-MNC labeled with SPION_NIRF-Rh_ homing and tracking by NIRF. In vivo NIRF images in the dorsal plane of control, young mice that received young cells (young → young), and young mice that received old cells (old → young) 2, 4, 24, and 72 h after cell transplantation. Color scale bar of the NIRF signal intensity (photons/s).

**Figure 8 biomedicines-09-00752-f008:**
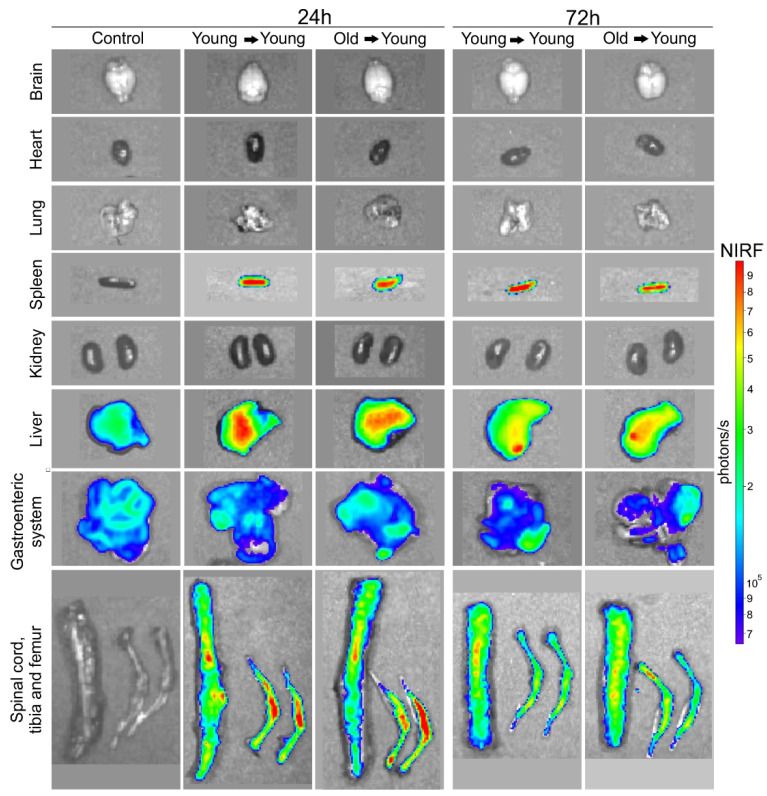
BM-MNC labeled with SPION_NIRF-Rh_ homing and tracking by NIRF. Ex vivo NIRF images of brain, heart, lung, spleen, kidneys, liver, gastroenteric system, spinal cord, tibia and femur of control, young mice that received young cells (young → young), and of young mice that received old cells (old → young) 24 and 72 h after cell transplantation.

**Figure 9 biomedicines-09-00752-f009:**
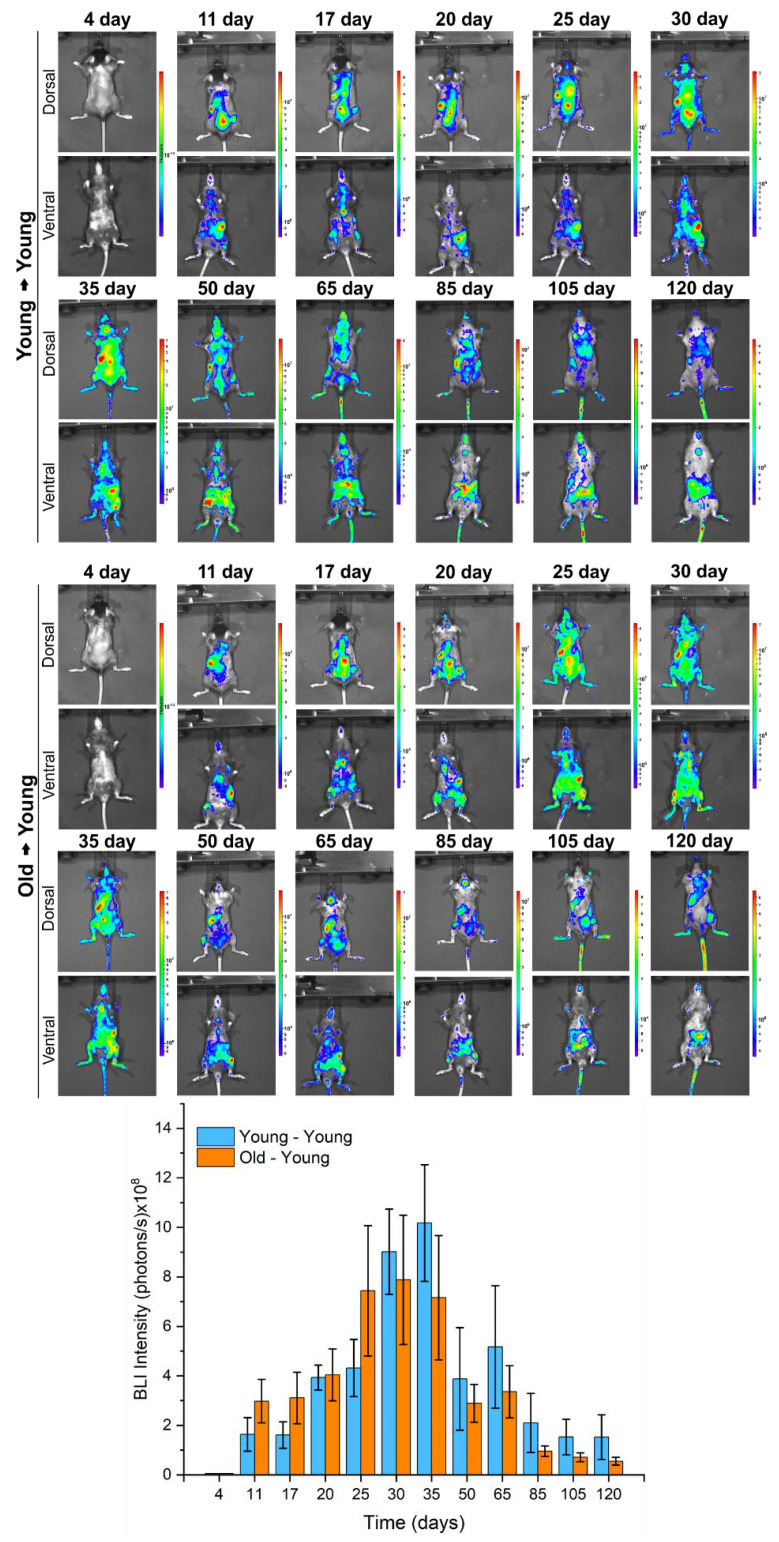
In vivo BLI images (dorsal and ventral planes) of young mice that received young cells (young → young) or old cells (old → young) over the course of 120 days (approximately each week in the first month and every 15 days within the next 3 months). Bar graph of BLI intensity of young mice that received young cells (blue bars) and young mice that received old cells (red bars) during 120 days of evaluation.

**Figure 10 biomedicines-09-00752-f010:**
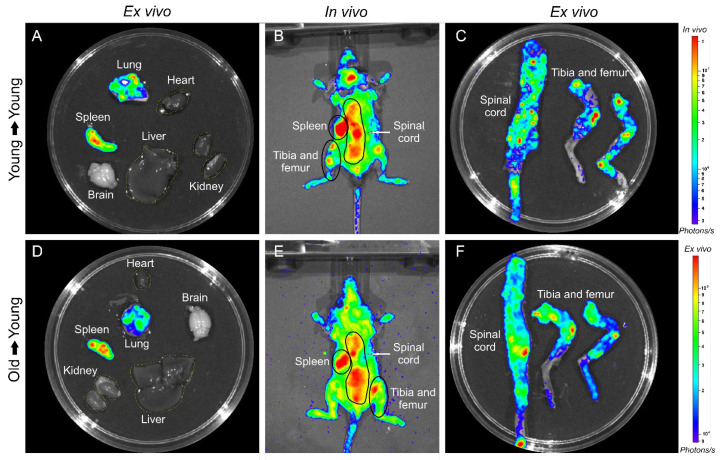
In vivo and ex vivo BLI images of young mice that receive young cells (**A**–**C**) or old cells (**D**–**F**), and images of their main organs (heart, spleen, lung, brain, kidney, liver, spinal cord, tibia, and femur) after 120 days of BMT.

**Figure 11 biomedicines-09-00752-f011:**
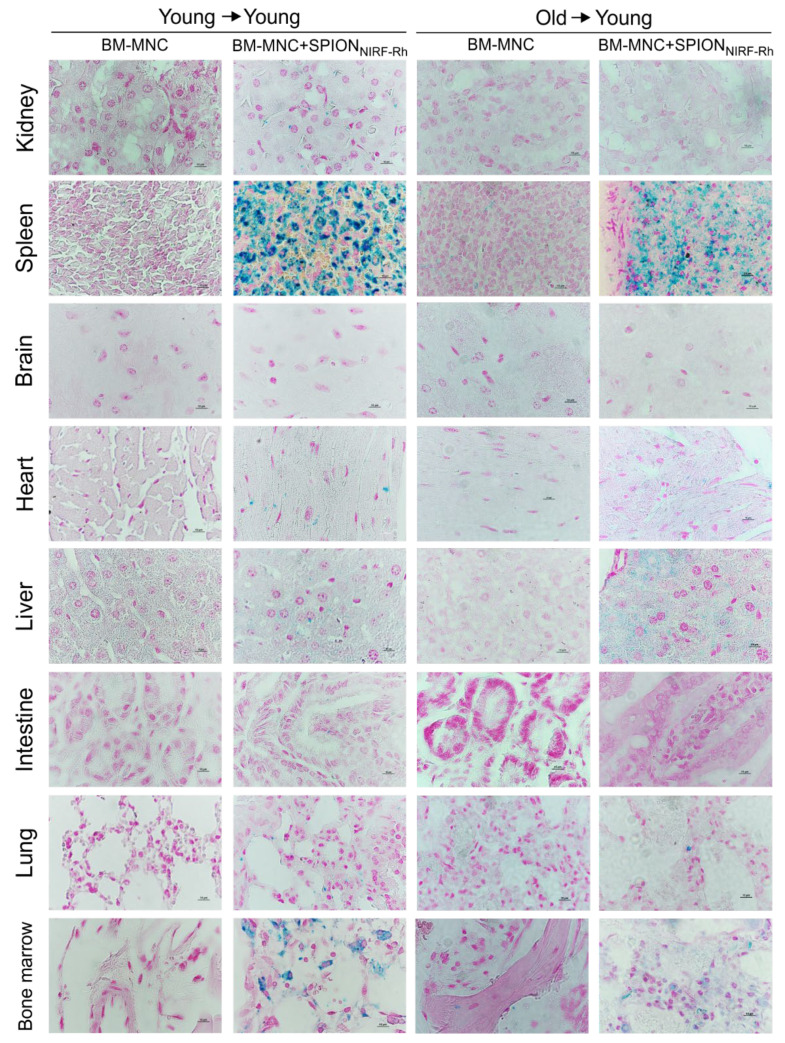
Histological analysis of the tissues of the main organs of animals submitted to BMT with young cells (young → young) and old cells (old → young), labeled and unlabeled with SPION_NIRF-Rh_, stained with Prussian blue and Nuclear fast red. Images magnified by 100×, and cut at 5 µm.

**Figure 12 biomedicines-09-00752-f012:**
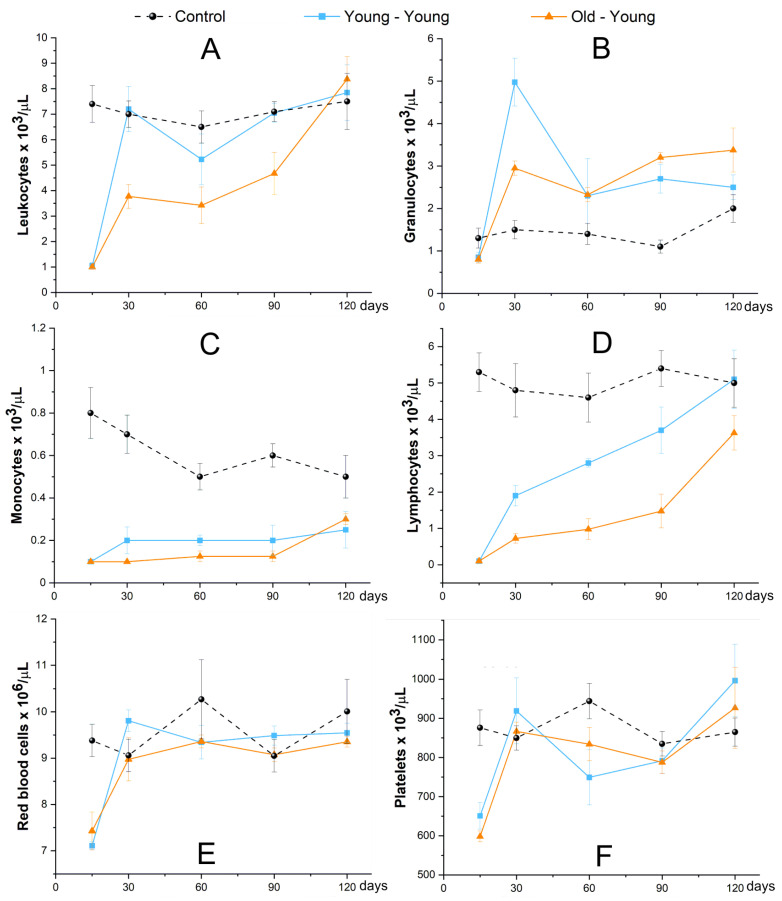
The hematological evaluation of animals that received cells of young donors (blue square scatter plot), or cells of old donors (red triangle scatter plot), and control without BMT (black circle scatter plot) for 120 days of evaluation of main blood components, such as: (**A**)—Leukocytes, (**B**)—Granulocytes, (**C**)—Monocytes, (**D**)—Lymphocytes, (**E**)—Red blood cells, and (**F**)—Platelets.

**Figure 13 biomedicines-09-00752-f013:**
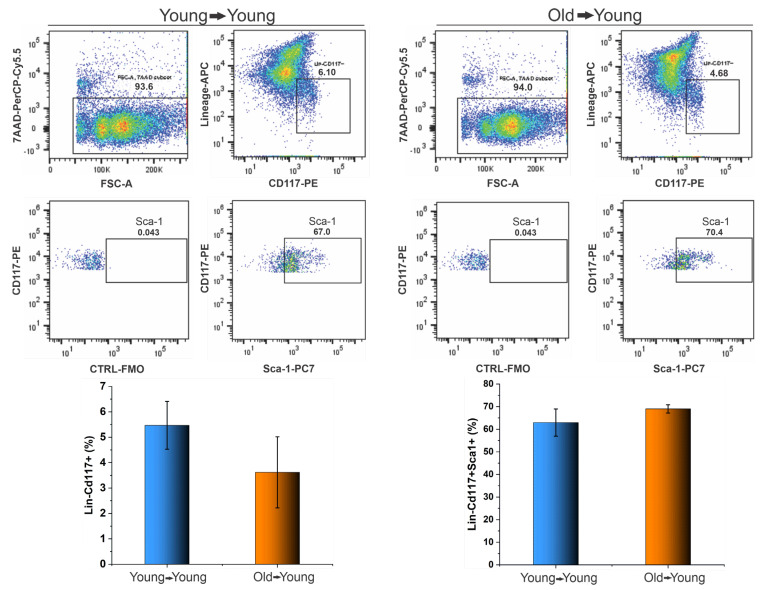
The quantification of LKS cells present in BM receptors after 120 days of BMT by flow cytometric analysis.

**Table 1 biomedicines-09-00752-t001:** BLI intensity signal of young and old BM-MNC labeled with different concentrations of SPION_NIRF-Rh_.

SPION_NIRF-Rh_ Concentrations (μg Fe/mL)	Mean ± SD × 10^8^ (Photons/s) of BLI Intensity Signal		N
	Young BM-MNC	Old BM-MNC	
0	1.378 ± 0.009	1.369 ± 0.001	10
10	1.360 ± 0.001	1.355 ± 0.007	10
30	1.343 ± 0.001	1.336 ± 0.009	10
50	1.325 ± 0.003	1.326 ± 0.003	10

SPION_NIRF-Rh_: multimodal nanoparticles; BM-MNC: bone marrow mononuclear cells; BLI: Bioluminescence.

**Table 2 biomedicines-09-00752-t002:** The iron mass and number of SPION_NIRF-Rh_ internalized per young and old BM-MNC labeled with 10, 30, and 50 µg of SPION_NIRF-Rh,_ determined using NIRF, ICP-MS, and MRI techniques.

[Fe] (µg/mL)	BM-MNC of Animal	NIRF		ICP-MS		MRI	
		Mass (pg Fe/Cell)	N° of SPION × 10^4^/Cell	Mass (pg Fe/Cell)	N° of SPION × 10^4^/Cell	Mass (pg Fe/Cell)	N° of SPION × 10^4^/Cell
10	Young	2.27 ± 0.09	2.85 ± 0.11	1.90 ± 0.10	2.39 ± 0.12	-	-
	Old	2.07 ± 0.07	2.61 ± 0.08	1.67 ± 0.06	2.10 ± 0.74	-	-
30	Young	3.06 ± 0.12	3.85 ± 0.15	3.21 ± 0.07	4.03 ± 0.93	-	-
	Old	3.02 ± 0.09	3.79 ± 0.11	3.01 ± 0.10	3.78 ± 0.13	-	-
50	Young	3.98 ± 0.16	5.00 ± 0.20	4.23 ± 0.09	5.32 ± 0.11	3.13 ± 0.24	3.93 ± 0.30
	Old	3.98 ± 0.14	5.00 ± 0.18	4.00 ± 0.07	5.03 ± 0.81	3.08 ± 0.17	3.87 ± 0.21

SPION_NIRF-Rh_: multimodal nanoparticles; BM-MNC: bone marrow mononuclear cells; NIRF: Near-infrared fluorescence; ICP-MS: Inductively Coupled Plasma Mass Spectrometry; MRI: Magnetic resonance imaging.

**Table 3 biomedicines-09-00752-t003:** BLI intensity values of young mice that received young cells or old cells for 120 days.

Time Evaluation (Days)	Mean ± Standard Deviation of BLI Intensity (Photons/s) × 10^8^	
	Young → Young	Old → Young
4	0	0
11	1.64 ± 0.68	2.98 ± 0.88
17	1.61 ± 0.53	3.11 ± 1.04
20	3.93 ± 0.50	4.04 ± 1.05
25	4.32 ± 1.15	7.44 ± 2.64
30	9.02 ± 1.72	7.88 ± 2.61
35	10.18 ± 2.35	7.16 ± 2.51
50	3.87 ± 2.07	2.89 ± 0.76
65	5.17 ± 2.47	3.36 ± 1.05
85	2.10 ± 1.19	0.96 ± 0.21
105	1.53 ± 0.71	0.71 ± 0.18
120	1.53 ± 0.90	0.55 ± 0.16

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
