# Peer review of "Multimodal Tracking of Hematopoietic Stem Cells from Young and Old Mice Labeled with Magnetic–Fluorescent Nanoparticles and Their Grafting by Bioluminescence in a Bone Marrow Transplant Model"

_biomedicines, 2021, doi:10.3390/biomedicines9070752_

Round 1
Reviewer 1 Report
The manuscript reports an innovative methods hematopoietic stem cell for homing and tracking. It’s well written and organized, although a selection should be made of the numerous References articles.
Some small changes should be taken into consideration:
- More intermediate date should by added for NIRF homing evaluation and HSC engraftment as continuous non invasive monitoring is the key aspect for this new evaluation method;
- Higher resolution image should be provide for all Figure
- Figures 3C, E, G should be clarified;
- Data in table 2 are redundant in the text (3.5.1. NIRF, 3.5.2. ICP-MS), they should be deleted and replaced with a critical comment;
- More time point should be added to Figure 7.
Author Response
Reviewer #1
The manuscript reports an innovative methods hematopoietic stem cell for homing and tracking. It’s well written and organized, although a selection should be made of the numerous References articles.
Some small changes should be taken into consideration:
- More intermediate date should by added for NIRF homing evaluation and HSC engraftment as continuous non invasive monitoring is the key aspect for this new evaluation method;
Answer: Thank you for your observation and suggestion. Our study adopted the NIRF homing evaluation of the transplanted cells in the following times of 2, 4, 24, and 72 hours based on the previous study,[1] that did not visualize the NIRF homing in the first moments after cell transplantation, since the NIRF technique has detection limitation due to the sensitivity affected by the cell distribution in the body (loss of the NIRF signal due to lack of cell concentration per area). For this reason, we decided to perform the NIRF image acquisition from 2 hours, and then 8, 24, and 72 hours after cell transplantation to visualize the cell homing dynamics. In addition, based on the literature results [1] that evaluated the NIRF homing before 2 hours (as 5, 15, 30 min ... ) until 24 hours, only from 3 hours, the NIRF homing results reported was compatible to the results obtained in the present study.
However, added more intermediate data in the NIRF homing evaluation in this study would be unfeasible, since the animals included in this study were elderly and for a new experiment, we would need to wait for the animals to age to be used as donors, which would take at least 18 months.
In the future, we intend to carry out this assessment at more time points using NIRF and with the aid of other techniques such as PET and MRI.
- Higher resolution image should be provide for all Figure
Answer: Thank you for your observation. High-resolution figures were inserted in the manuscript as requested.
- Figures 3C, E, G should be clarified;
Answer: Thank you for your suggestion. We improved the description of the results as requested, clarifying the Figure 3C, E, G information in the manuscript.
- Data in table 2 are redundant in the text (3.5.1. NIRF, 3.5.2. ICP-MS), they should be deleted and replaced with a critical comment;
Answer: Thank you for your observation. We modified the text mentioned by the reviewer, reducing and condensing only the most relevant data on quantification.
- More time point should be added to Figure 7.
Answer: Thank you for your suggestion. However, we performed the NIRF cell homing evaluation until 96 hours, but in the last time point (96 h) did not find any NIRF signal in the evaluated animals. A similar temporal evaluation pattern of the NIRF homing of our study was reported, and the last time point at 72 hours showed a great NIRF signal reduction in the region of interest [1]. In addition, homing is a process that in general occurs in the first 24 hours [2,3], based on this information we opt to show the cell homing until 72 hours.
REFERENCE
- Ushiki, T.; Kizaka-Kondoh, S.; Ashihara, E.; Tanaka, S.; Masuko, M.; Hirai, H.; Kimura, S.; Aizawa, Y.; Maekawa, T.; Hiraoka, M. Noninvasive tracking of donor cell homing by near-infrared fluorescence imaging shortly after bone marrow transplantation. PloS one 2010, 5, e11114, doi:10.1371/journal.pone.0011114.
- Lapidot, T.; Dar, A.; Kollet, O. How do stem cells find their way home? Blood 2005, 106, 1901-1910, doi:10.1182/blood-2005-04-1417.
- Xie, Y.; Yin, T.; Wiegraebe, W.; He, X.C.; Miller, D.; Stark, D.; Perko, K.; Alexander, R.; Schwartz, J.; Grindley, J.C., et al. Detection of functional haematopoietic stem cell niche using real-time imaging. Nature 2009, 457, 97-101, doi:10.1038/nature07639.

Reviewer 2 Report
Manuscript submitted by Olivera et. al "Multimodal tracking of hematopoietic stem cells from young and old mice labeled with magnetic-fluorescence nanoparticles and their grafting by bioluminescence in a bone marrow transplant model" in Biomedicines (biomedicines-1246360). The presented data will be important in terms of bioengineering achievement as well as the possibility of application of fluorescent dye as a biosensor. In this regard, this paper is worth for publication in the journal.
Author Response
Reviewer #2
Manuscript submitted by Olivera et. al "Multimodal tracking of hematopoietic stem cells from young and old mice labeled with magnetic-fluorescence nanoparticles and their grafting by bioluminescence in a bone marrow transplant model" in Biomedicines (biomedicines-1246360). The presented data will be important in terms of bioengineering achievement as well as the possibility of application of fluorescent dye as a biosensor. In this regard, this paper is worth for publication in the journal.
Answer: We would like to thank you for your considerations and your time dedicated to reviewing our manuscript.

Reviewer 3 Report
The authors have done a lot to establish a method to evaluate homing and tracking of implanted stem cells. Their method would be useful to appreciate the appropriateness of the donor cells.
Author Response
Reviewer #3
The authors have done a lot to establish a method to evaluate homing and tracking of implanted stem cells. Their method would be useful to appreciate the appropriateness of the donor cells.
Answer: We would like to thank you for your considerations and your time dedicated to reviewing our manuscript.
